# Improvement of Erosion-Corrosion Behavior of AISI 420 Stainless Steel by Ion-Assisted Deposition ZrN Coatings

**Yuntao Xi [1,2], Lin Wan [1], Jungang Hou [3], Zhiyong Wang [3], Lei Wang [1,4,5,*], Daoyong Yang [2,*], Daoxin Liu [6] and Shubin Lei [1]**

[1] School of Material Science and Engineering, Xi'an Shiyou University, Xi'an 710065, China; ytxi@xsyu.edu.cn (Y.X.); curry30linfor@gmail.com (L.W.); shubinlei@xsyu.edu.cn (S.L.)
[2] Petroleum Systems Engineering, Faculty of Engineering and Applied Science, University of Regina, Regina, SK S4S 0A2, Canada
[3] No.1 Oil Production Plant of Changqing Oilfield Company CNPC, Xi'an 716002, China; lyd1135436906@gmail.com (J.H.); 951216zxw@gmail.com (Z.W.)
[4] State Key Laboratory of Metastable Materials Science and Technology, Yanshan University, Qinhuangdao 066004, China
[5] State Key Laboratory of Advanced Metals and Materials, University of Science and Technology Beijing, Beijing 100083, China
[6] Institute of Corrosion and Protection, School of Aeronautics, Northwestern Polytechnical University, Xi'an 710072, China; liudaox@nwpu.edu.cn
[*] Correspondence: wanglei@xsyu.edu.cn (L.W.); tony.yang@uregina.ca (D.Y.); Tel.: +86-157-7192-1910 (L.W.); +1-306-337-2660 (D.Y.)

**Abstract:** In this paper, a pragmatic technique has been developed to evaluate the erosion-corrosion behavior of three kinds of ZrN coatings (i.e., monolayer, multilayer, and gradient layers) which were deposited on AISI 420 martensitic stainless steel using an ion-assisted deposition technology. Among them, the monolayer coating refers to the coating with no change in composition and structure, the multilayer coating refers to the coating with alternating change of Zr/ZrN, and the gradient coating refers to the ZrN coating by increasing $N_2$ partial pressure gradually. The morphology, composition, and microhardness of these ZrN coatings were examined by means of integrating the scanning electron microscopy (SEM), X-ray diffraction (XRD), and Knoop hardness measurements, while anodic polarization tests and salt fog spray tests in a simulated industrial environment have been performed to evaluate and identify the corrosion mechanisms of these coatings. The surface microhardness and corrosion resistance of the AISI420 martensitic stainless steel is found to be significantly improved by depositing the ion-assisted deposition ZrN coatings. The study indicates that the erosion-corrosion behavior in the slurry is the result of the synergistic effect of small-angle erosion and acid solution corrosion. Three ZrN coatings hinder the slurry erosion-corrosion behavior from two aspects (i.e., erosion resistance of small-angle particles as well as corrosion resistance of the substrate), thereby significantly improving the erosion-corrosion resistance of AISI 420 stainless steel. In addition, the ZrN gradient coatings show a much better erosion-corrosion resistance than that of the monolayer/multilayer ZrN coating because they have excellent crack resistance, bearing capacity, and electrochemical performance.

**Keywords:** ion-assisted deposition; ZrN coatings; erosion-corrosion; stainless steel

## 1. Introduction

The martensitic stainless steel AISI 420 has been widely used in manufacturing turbine blades because of its excellent mechanical properties and high corrosion resistance. However, due to the low hardness and poor wear resistance, AISI 420 martensitic stainless blades are subject to solid-liquid two-phase erosion-corrosion [1,2]. As for a steam turbine, its erosion-corrosion remains a crucial problem, especially on the last blades at its tip end. When the steam turbine is working, the surfaces of these blades are impacted by high-speed

airflow containing a small amount of liquid drops and solid particles, leading to the fact that the surfaces of these blades are gradually eroded and peeled off. In this way, the blade strength is significantly reduced, aerodynamics of the cascade is worsened, and there may be even sudden blade breakage, posing a serious threat to the safe operation of the steam turbine [3,4].

Numerous efforts have been made to improve the erosion-corrosion performance of steam turbine blades. The gas metal arc welding or high velocity oxygen fuel (HVOF) CaviTec coatings can improve the surface erosion-corrosion performance, but they are prone to micro-cracks and cavitation corrosion because of their hard and brittle surface [5,6]. Nagentrau et al. [7] and Kuznetsov et al. [8] improved the erosion resistance of blades with overlaying welding Tungsten carbide coatings and laser cladding Stellite 6 alloy (alloys based on Co, 150 μm); however, it is easy to cause deformation defects because such treatments undergo large heat inputs. At present, the aforementioned improvements are found to be only suitable for small turbines [5,7,8]. Recently, the ion-assisted deposition technology allows us to treat a surface by combining traditional physical vapor deposition (e.g., multi-arc ion plating and ion sputtering) with various ion-assisted sources [9,10]. Such combined treatment improves the film performance at a low deposition temperature and ensures that the blades do not deform due to the low heat input. Compared with physical vapor deposition (PVD), such combined treatment leads to better coating bonding strength, denseness, and deposition atom mobility [11–13].

In recent years, some progress has been made in improving the small-angle solid particle erosion properties of ZrN coatings prepared by ion-assisted deposition technology [14–16]. However, there are few studies on the solid-liquid two-phase erosion-corrosion resistance. When the turbine blades rotate at a high speed, they are not only eroded by solid particles, but also corroded by the surrounding medium, which is a synergistic effect of the two aspects [6,8,17,18]. Recent studies have shown that multilayer films and gradient films have greater advantages than single-layer films in improving the corrosion resistance of metal materials, but there are fewer studies on solid-liquid two-phase erosion-corrosion [19–21]. So it is extremely important to evaluate and identify the mechanisms of corrosion resistance and erosion-corrosion behaviors of ZrN coatings with different structures.

In this work, a pragmatic technique has been developed to evaluate the erosion-corrosion behavior for various ZrN coatings (i.e., monolayer, multilayer, and gradient layers). The ZrN coatings were prepared on the surface of AISI 420 martensitic stainless steel by ion-assisted arc deposition technology. Then, SEM, XRD, and Knoop hardness tests were performed to evaluate morphology, composition, and microhardness of the coatings, while salt fog spray corrosion tests and modified rotary slurry erosion-corrosion tests were conducted to measure the electrochemical potentials and erosion-corrosion volume damage ratio. Moreover, effect of the coating structure on the erosion-corrosion performance has been examined and analyzed.

## 2. Experiments

### 2.1. Materials

The chemical composition (wt%) of the substrate material AISI 420 martensitic stainless steel is presented in Table 1. The material was annealed at 860 °C for 4 h, followed by oil quenching at 900 °C for 3 h and tempering at 600 °C for 6 h. Disk-type samples of 30 mm in diameter and 10 mm in thickness were cut from a stainless steel bar. Prior to surface treatments, the sample surfaces were ground and polished with 400#, 800#, and 1200# abrasive papers. Finally, the surfaces were ultrasonically washed with acetone and distilled water, and dried for ion-assisted arc deposition.

**Table 1.** Chemical composition of AISI 420 martensitic stainless steel (wt%).

| C | Cr | Mn | Si | Ni | Cu | P | S | Fe |
|---|---|---|---|---|---|---|---|---|
| 0.190 | 12.650 | 0.200 | 0.280 | 0.120 | 0.110 | 0.028 | 0.007 | balance |

## 2.2. Experimental Setup

Figure 1 shows a schematic diagram and detailed photographs of the PIEMAD-03 (Foxin Vacuum Technology Co., Ltd., Foshan, China) ion-assisted multi-arc deposition system which combines the traditional multi-arc ion plating with ion-assisted enhancement methods. The system consists of a slit plane ion-assisted source, a multi-arc ion zirconium target, a pulsed bias power supply and a vacuum system.

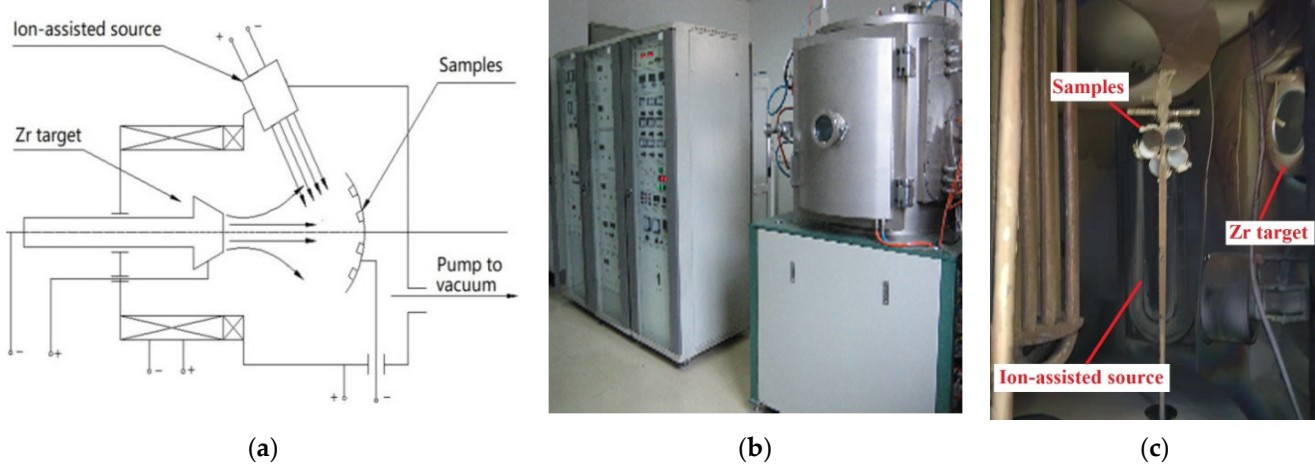

**Figure 1.** PIEMAD-03 ion-assisted multi-arc deposition system: (**a**) Schematic diagram, (**b**) photos of instruments, (**c**) corresponding components.

The slot plane ion source is an important ion-assisted enhancement deposition device, PIEMAD-03 (Foxin Vacuum Technology Co., Ltd., Foshan, China), in which the working gas was high-purity argon. Before and during the preparation of coatings, the slit plane ion source can ionize a lot of Ar$^+$, and bombard the surface of the sample to achieve the purposes of cleaning, film/base atom mixing, and enhancing the deposition. Because of the enhanced deposition effect of the slit plane ion-assisted source, the distance between the sample and the target reaches 150 mm, which can significantly reduce the temperature of the sample below 300 °C. The zirconium target (99.95 wt%) was bombarded either by Argon ion in the case of pure Zr layers or nitrogen ion in the case of ZrN layers. The pulsed bias power (Haoyuan, Guangzhou, China) supply is used to bias the sample, and increase the film/base bond strength and deposition rate. During the deposition process, a bias voltage of −350 V was applied to the substrate. The vacuum system is a two-stage pump (i.e., molecular pump-mechanical pump) vacuum system, which can maintain a pressure of 5–8 × 10$^{-1}$ Pa in the vacuum chamber to prevent the coating from being oxidized.

Figure 2 shows a modified rotary erosion-corrosion test device (home-made) that consists of a glass container, a polyethylene sample holder, a glass shaft, a governor device, and a main motor. The glass container is used to store the experimental slurry, while the prime motor is used to provide liquid rotation power. The polyethylene sample holder is used to hold and seal the sample, and only the test surface is exposed in the solution. The governor device is mainly used to control the erosion-corrosion linear velocity of the sample.

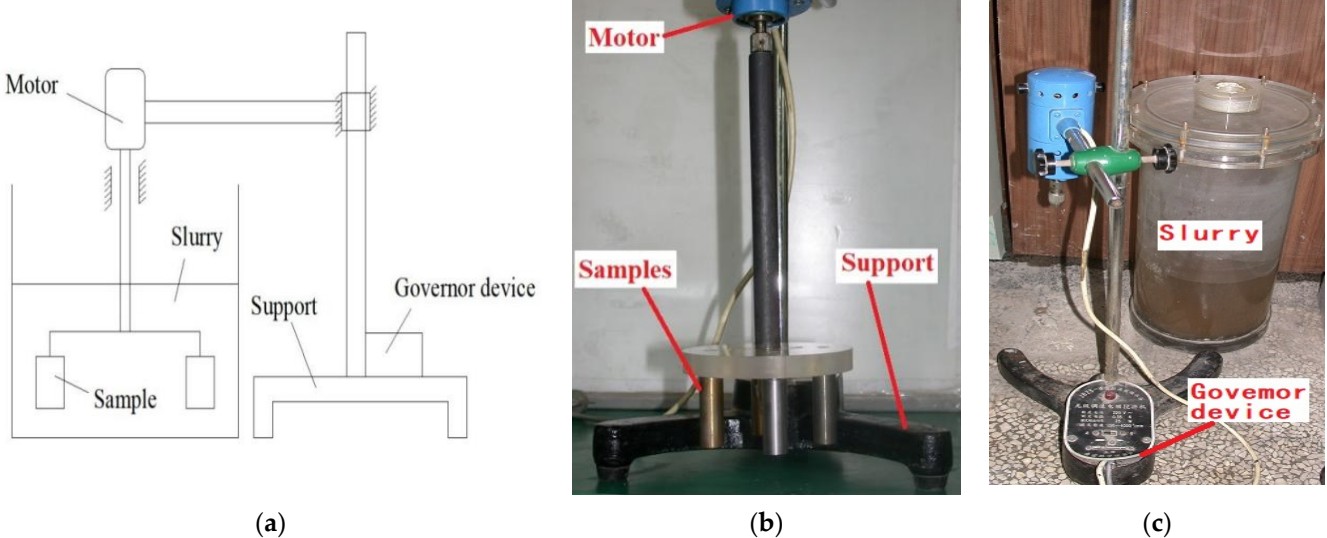

|     |     |     |
| :-: | :-: | :-: |
| (**a**) | (**b**) | (**c**) |

**Figure 2.** (**a**): Schematic diagram, (**b**,**c**): corresponding components of the testing corrosion-erosion setup.

### 2.3. Experimental Procedures

### 2.3.1. Coating Deposition

Three kinds of ZrN coatings were prepared using the PIEMAD-03 ion-assisted multi-arc ion deposition equipment. Each coating contains a 2.0 μm Zr primer layer. Coating structures are designed as follows: (1) ZrN monolayer coating (i.e., Coating #1) is composed of a 6.0 μm ZrN surface layer; (2) ZrN multilayer coating (i.e., Coating #2) is composed of 5 layers of Zr/ZrN surface alternate deposition layers, in which thickness of each surface layer is 0.9–1.2 μm and the total thickness is 8.0 μm; and (3) ZrN gradient coating (i.e., Coating #3) consists of a 6.0 μm ZrN gradient surface layer, where the high-purity $N_2$ partial pressure is gradually increased from 0.1 Pa to 4.0 Pa during the preparation of the gradient surface coating.

### 2.3.2. Coating Characterization

The surface cross-section morphology of the ZrN coatings was examined with an HITACHIS-570 scanning electron microscope (HITACHI, Tokyo, Japan) operated at 20 kV. The surface compositions of these coatings were analyzed with an X-ray diffraction facility (X'Pert PRO, PANALYTICAL, Amsterdam, the Netherlands). The surface microhardness profiles were measured by means of an HV-1000 Knoop hardness tester (BIUGED, Guangzhou, China) for 20 s with a Knoop pressure head [4]. In the hardness test, each point is tested five times. The hardness test error is given, and the average value is taken as the final result.

### 2.3.3. Corrosion Tests

The electrochemical corrosion behavior of samples was tested using a PARSTAT2273 electrochemical measuring system (AMETEK, Santiago, California, USA) with a three-electrode flat cell. The scanning potential was in the range of −0.6 to +0.8 V, and the potential scan rate was 0.166 V/s. The test solution (pH = 3 ± 0.2) consisted of 5.0 wt% NaCl and $H_2SO_3$ solution maintained at 25 ± 2 °C in order to simulate the service environment of turbine blades [22].

The salt fog spray corrosion test was performed using a SY/Q-750 salt fog spray corrosion test instrument (Aoke, Wuxi, China) in accordance with ASTM B117 [2]. The solution used for salt fog spray corrosion tests was the same as that of the electrochemical test. The salt fog spray test was conducted continuously with 20 mL/min $CO_2$ gas flow. The test temperature was maintained at 35 ± 2 °C, and the relative humidity was maintained at 94 ± 4% for 720 h.

2.3.4. Slurry Erosion-Corrosion Tests

The slurry erosion corrosion experiment was carried out on a rotary testing machine (Figure 2). The solution used for slurry erosion-corrosion tests was the same as that of the electrochemical test. In order to simulate the erosion-corrosion environment of industrial blades, a content of 10 wt% silica sand was added to the erosion-corrosion solution. The total slurry volume of the erosion-corrosion tests is 10 L. The silica sand was used as an abrasive agent with a particle size between 150 μm and 250 μm, and suspended in the solution. The erosion-corrosion line speed was 4.8 m/s, and the test time was 24 h. Since the density of the substrate and each coating was different, the erosion process was evaluated according to the volume loss. In order to compare the volume loss caused by the erosion process, the profiler was used to measure the erosion pit morphology of the sample after the erosion test, and the erosion volume loss was calculated to evaluate the corrosion process. The sample is cleaned in alcohol by ultrasonic before weighing so that the influence of impurities can be avoided. Electrochemical tests, salt spray corrosion tests, and erosion-corrosion tests were carried out three times for each coating to ensure the consistency of the experimental results.

## 3. Results

*3.1. Surface and Cross-Sectional Morphology*

The photograph of particle-assisted deposition ZRN coating sample is shown in Figure 3. The surface micrographs of the ion-assisted arc deposition ZrN coatings are shown in Figure 4. The coating is found to be smooth and golden in color. There are many small particles of bright white color on the surface of the coating, as shown in secondary electron image (SE, Figure 4a), which can be more clearly observed with the scanning backscattering electron microscope (BSE). As can be observed in Figure 4b, the micro-convex particles on the surface of the ZrN coating and the circular pits are left after individual particles fall off. This phenomenon is caused by the incompletely ionized zirconium droplets sputtered from the arc spot on the cathode arc target [23].

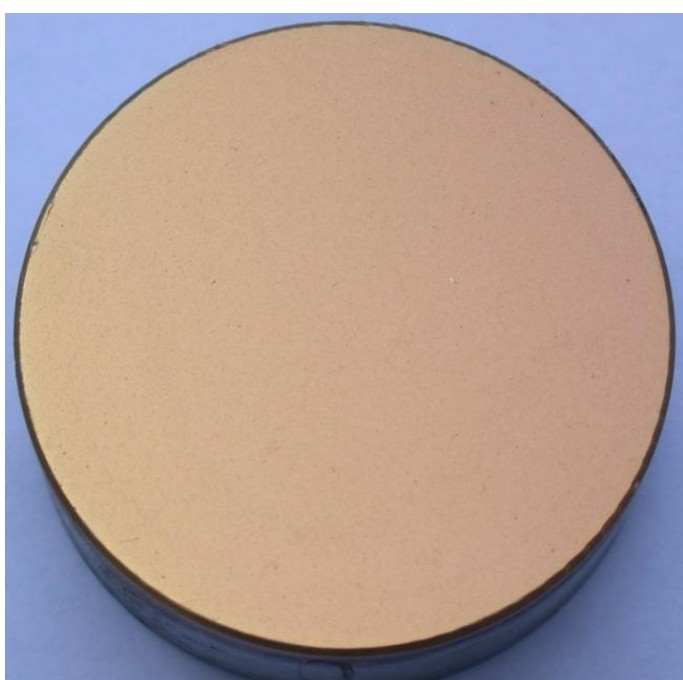

**Figure 3.** The photograph of ion-assisted arc deposition ZrN coating sample.

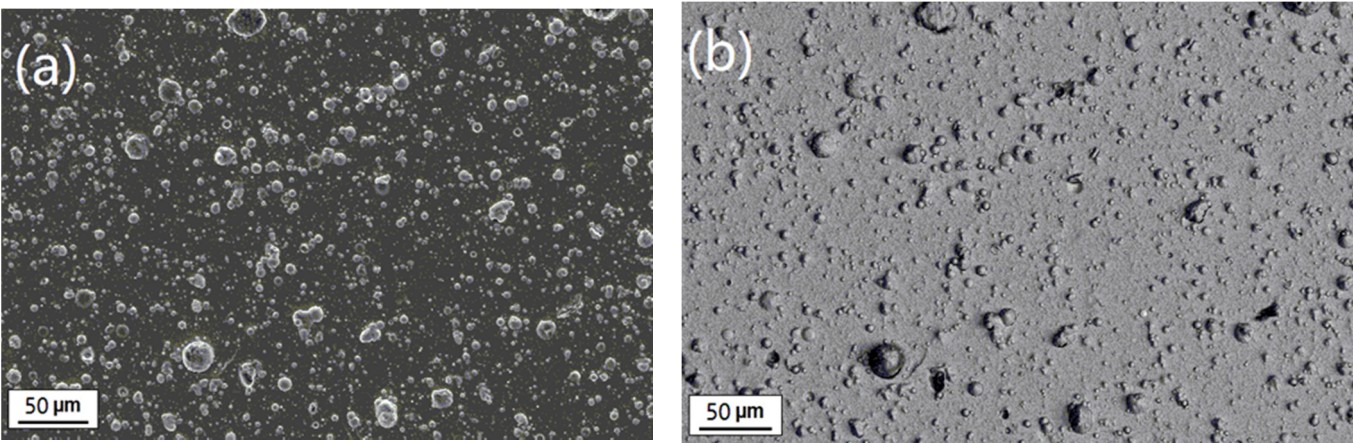

**Figure 4.** Surface micrographs of the ion-assisted arc deposition ZrN coatings: (**a**) SE and (**b**) BSE.

Figure 5a shows the cross-sectional morphology of Coating #1 which is composed of a Zr underlayer with a thickness of 2.0 μm and a ZrN single homogeneous surface layer (6.0 μm). There is a clear interface between the underlayer and the surface layer. As shown in Figure 5b (Coating #2: A Zr underlayer of 2.0 μm and 5 Zr/ZrNs alternately deposited layers), each alternately deposited layer (1.2 μm) looks very dense and alternate in color. Figure 5c indicates that Coating #3 consists of a Zr underlayer (2.0 μm) and a ZrN gradient layer (6.0 μm). Compared with Coatings #1 and #2, the interface of Coating #3 between the underlayer and the surface layer is less obvious. Because of the strong ion bombardment of sample surface by the slit plane ion source, the bonding strength between the layers is significantly increased. As such, it can be seen form Figure 5, all of the three ZrN coatings deposited by the ion-assisted deposition technology are very dense, defect-free, and tightly bonded. This observation is similar to those documented elsewhere [11,24].

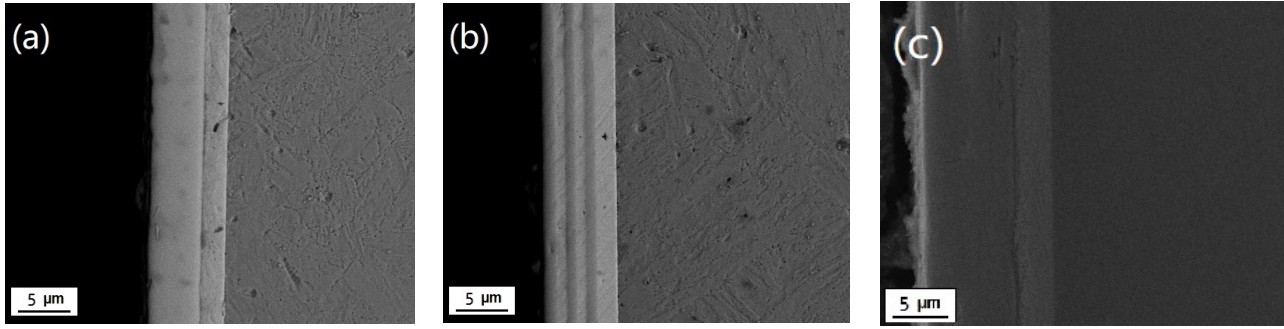

**Figure 5.** Cross-sectional micrograph of ZrN coatings with three structures. (**a**) Coating #1 single layer, (**b**) Coating #2 multilayer, (**c**) Coating #3 gradient layer.

Figure 6 shows the element distribution curve of Coating #1 and Coating #3 along the layer-depth. It can be seen that in the Coating #1, the Zr layer is located between the ZrN layer and the stainless steel substrate. Similarly, it can be seen from Coating #3 that there is also a Zr layer between the ZrN gradient film and the stainless steel substrate. Comparing Coating #1 and Coating #3, it can be seen that the changes of Zr and N in the ZrN gradient film layer are smoother and slower than that of the single layer, and there is no obvious step. Therefore, it can be seen from the element change diagram that the Coating #3 is a gradient film layer.

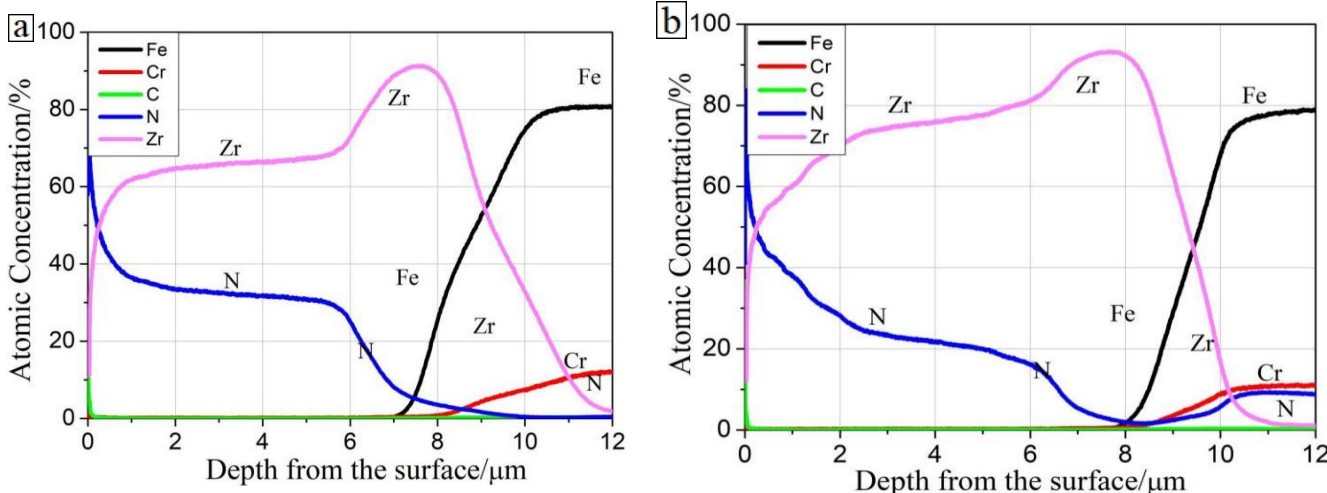

**Figure 6.** Element distribution curve vs. layer-depth (atom percentage). (**a**) Coating #1 single layer, (**b**) Coating #3 gradient layer.

Because the surface composition of the three coatings is the same, take Coating #1 as an example for analysis. According to the XRD test results (see Figure 7). The surfaces of the three ZrN coatings are confirmed to be pure ZrN, and exhibit a preferred orientation of the (200) plane. This is the same as those observed by Jiménez et al. [25].

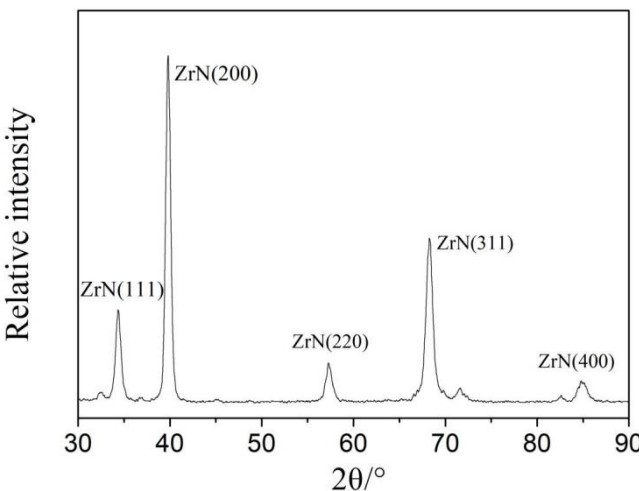

**Figure 7.** XRD pattern of ZrN coatings.

### 3.2. Microhardness

The microhardness of AISI420 stainless steel with three kinds of ZrN coating samples were measured under several test loads (Figure 8). The microhardness of the substrate is 278 $HK_{0.1}$, while that of the ZrN coatings are higher than 3000 $HK_{0.1}$. The hardness of substrate is relatively low and basically does not change with loads. The microhardness of Coating #1 was the highest of about 3737 $HK_{0.1}$ among the three ZrN coatings, and the hardness of Coating #2 was the lowest of about 3074 $HK_{0.1}$. The surface microhardness of Coating #3 was 3381 $HK_{0.1}$ that is decreased more gradually with the increase of the load, while the surface hardness of the other two coating samples is decreased significantly with the increase of the load.

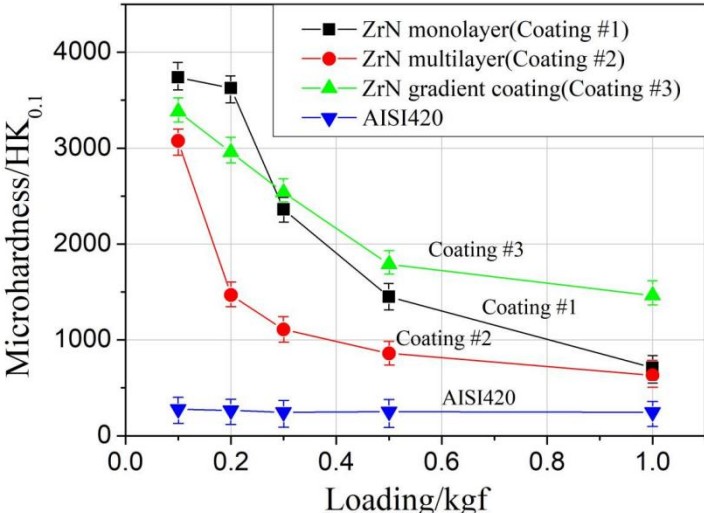

**Figure 8.** Surface microhardness profiles as a function of test loading.

When the test load is smaller, the ZrN single-phase homogeneous surface layer of Coating #1 is the thickest and has the highest bearing capacity, and thus the apparent microhardness is the highest. When the test load is increased to a certain value, the coating is broken and the microhardness is decreased rapidly [26]. Coating #2 has a multilayer in which Zr metal layer and ZrN ceramic layer are alternatively deposited. Since the Zr layer and the ZrN layer are both very thin and the hardness of the Zr layer is low, the comprehensive bearing capacity and surface microhardness of Coating #2 are the lowest, and they decrease rapidly with an increasing load [27]. Coating #3 has a gradient structure with gradually changing coating composition from the inner Zr metal layer to the outer ZrN ceramic layer. As shown in Figure 8, when the test load is gradually increased, the micro hardness of Coating #3 changes linearly and decreases slowly. It is also shown that Coating #3 has better deformation coordination ability, crack resistance, and bearing capacity. These scientific findings are consistent with those documented elsewhere [28].

### 3.3. Corrosion Resistance

### 3.3.1. Salt Fog Spray Tests

The corroded surface morphology of the AISI420 stainless steel samples and three ZrN coating samples after 720 h salt fog testing is illustrated in Figure 9. The red rust can be seen on the substrate sample, which has poor corrosion resistance (Figure 9a) [29,30]; however, there is almost no red rust on the surfaces of the three coating samples. As can be seen in Figure 9b–d, the ZrN monolayer, multilayer, and gradient layers have significantly improved the corrosion resistance of AISI420 stainless steel substrate.

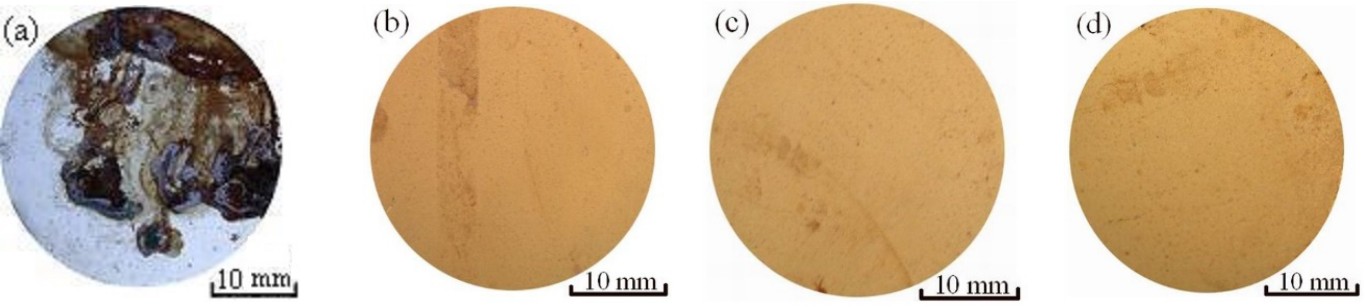

**Figure 9.** Corroded surface morphology of different samples. (**a**) AISI420, (**b**) Coating #1, (**c**) Coating #2, (**d**) Coating #3 after 720 h salt fog spray tests.

### 3.3.2. Electrochemical Behavior Tests

Figure 10 shows the anodic polarization curves of AISI420 stainless steel and three ZrN coating samples in 5.0 wt% NaCl + $H_2SO_3$ solution (pH = $3 \pm 0.2$). As can be seen from Figure 10, the AISI420 stainless steel sample does not have an evident passivation region, and the corrosion potential ($E_{corr}$) is $-0.580$ V. However, the corrosion potentials ($E_{corr}$) of the Coatings #1, #2, and #3 samples are $-0.272$ V, $-0.238$ V, and $-0.273$ V, respectively, while the polarization curves show evident passivation regions. According to the comparison of the corrosion potential ($E_{corr}$), the third ZrN coating samples exhibit a much higher corrosion resistance. According to the Tafel's linear extrapolation method [31,32], the corrosion current density ($I_{corr}$) of the AISI420 substrate is $3.77 \times 10^{-6}$ A/cm$^2$. On the other hand, the corrosion current densities ($I_{corr}$) of Coatings #1, #2, and #3 samples are $6.54 \times 10^{-7}$ A/cm$^2$, $7.32 \times 10^{-8}$ A/cm$^2$, and $1.58 \times 10^{-7}$ A/cm$^2$, respectively. Compared with the corrosion current of the AISI420 substrate, the ZrN monolayer is reduced by 82.7% under the same corrosion conditions, while the ZrN multilayer and the gradient layer are reduced by 98.1% and 95.8%, respectively.

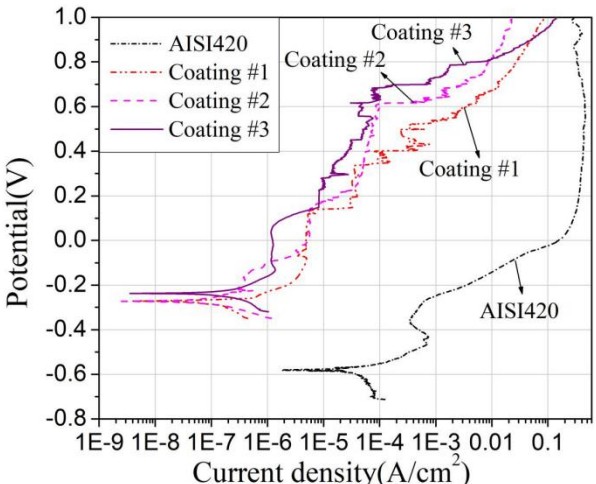

**Figure 10.** Polarization curves of different samples in 5.0 wt% NaCl + $H_2SO_3$ solution (pH = $3 \pm 0.2$).

### 3.3.3. Erosion-Corrosion Behavior

Table 2 shows the results of erosion-corrosion tests of AISI420 steel and three kinds of structure ZrN coating samples in an acidic slurry solution. Since the density of the substrate and each coating was different, the erosion process was evaluated according to the volume loss. In order to compare the volume loss caused by the erosion process, the profiler was used to measure the erosion pit morphology of the sample after the erosion test, and the erosion volume loss was calculated to evaluate the corrosion process. Volume loss refers to the volume reduction (in mm$^3$) after erosion-corrosion test. The volume damage ratio refers to the proportion of volume loss of coated samples in uncoated samples during erosion-corrosion test. If the volume loss of the untreated sample is positioned at 100%, the volume loss of the three surface treated samples after the 24 h slurry erosion corrosion test will be 6.2, 6.5, and 4.7%, respectively. This means that all three coating structures, i.e., ZrN monolayer, multilayer, and gradient layer have significantly improved the solid-liquid two-phase erosion-corrosion resistance of the AISI420 stainless steel.

**Table 2.** Erosion-corrosion results of AISI420 steel with and without coatings.

| Test Medium | Sample | Volume Loss (mm³) | Erosion-Corrosion Volume Damage Ratio (%) |
|---|---|---|---|
| Acid slurry (pH: 3 ± 0.2) | AISI 420 | 21.5 | - |
| | ZrN monolayer(Coating #1) | 1.34 | 6.2 |
| | Zr/ZrN multilayer(Coating #2) | 1.40 | 6.5 |
| | ZrN gradient layer(Coating #3) | 1.02 | 4.7 |

Figure 11 shows the surface micrographs of samples after slurry erosion-corrosion tests. It can be seen from the figure that, under the action of scouring, deep plough grooves appeared on the surface of AISI 420 sample. The erosion-corrosion damage of the substrate is serious. However, only shallow scratches appeared on the surface of the three coating samples. Compared with the plough groove caused by the uncoated sample, the number of scratches on the coated sample is less and the depth is shallow. The erosion-corrosion resistance was significantly improved through the PVD surface treatment method.

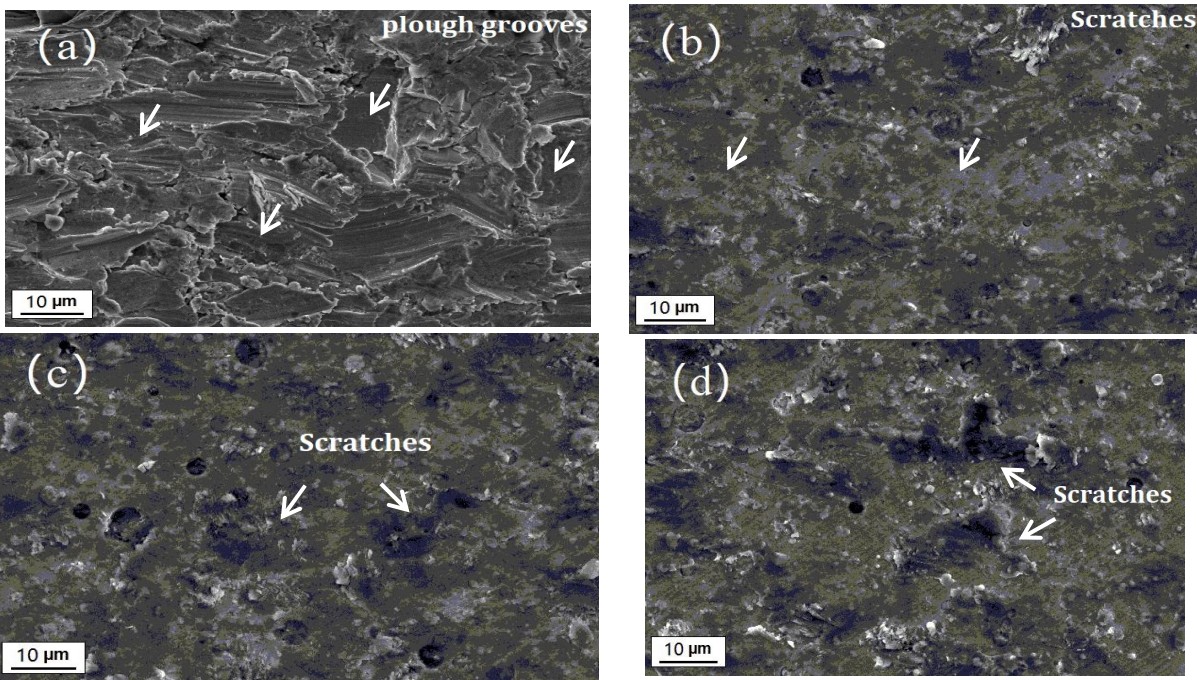

**Figure 11.** SEM surface micrographs of samples after slurry erosion-corrosion tests. (**a**) AISI 420, (**b**) Coating #1 single layer, (**c**) Coating #2 multilayer, (**d**) Coating #3 gradient layer.

## 4. Discussion

It can be seen that the ZrN coatings with three structures can significantly improve the corrosion resistance of AISI420 stainless steel, and the improvement effects of Coatings #2 and #3 are more significant. These phenomena can be explained with the following three main reasons. First, as cross-sectional morphology presented in Figure 4, the three ZrN coatings can effectively prevent the corrosive medium from contacting the substrate as they are relatively dense and have a certain thickness [33]. Second, during the preparation of the ZrN monolayer, micro-pores and other defects will be formed in the layer due to the existence of the columnar crystal structure of the layer and the surface droplets [27,34]. During the preparation of ZrN multilayer and gradient layer, however, changes in the partial pressure of $N_2$ will cause changes in the composition of the film layer, resulting in micro-pores closure and limiting their growth. Furthermore, in electrochemical behavior test, new micro-pores will be formed at other suitable locations, so that there will be

no channels or cracks from the surface of the film layer to the inner substrate, which prevents the substrate from directly contacting the electrolyte solution [35]. Therefore, both the ZrN multilayer and gradient layer films have better electrochemical corrosion protection effect on the AISI420 substrate than the ZrN monolayer coating. Finally, the formation of columnar crystals and micro-pores in the film layer can be better prevented when the Zr metal layer and the ZrN ceramic layer are alternately deposited [15,21]. In addition, the Zr metal layer has an excellent corrosion resistance, which ultimately makes the electrochemical corrosion resistance of the ZrN multilayer coating is the best among the three coatings.

In slurry erosion-corrosion tests, as shown in Table 2, ZrN monolayer, multilayer, and gradient layer have significantly improved the solid-liquid two-phase erosion-corrosion resistance of the AISI420 stainless steel, especially multilayer, and gradient layer. Figure 12 shows the surface profiles and curves of slurry erosion-corrosion pits of samples after erosion-corrosion tests. It can be seen that erosion-corrosion pits of AISI 420 are deeper, and volume loss of pits of three kinds of surface treated samples is lower than that of AISI 420. At a 30-degree erosion-corrosion angle, the erosion-corrosion mechanism of the material is the result of the combined effect of micro-cutting and corrosion. The erosion-corrosion resistance mainly depends on the surface hardness and corrosion resistance of the material. Because the substrate of AISI 420 sample has low hardness and poor wear resistance, the bulk loss is the largest. After three kinds of surface treatments, the surface hardness of the sample was significantly improved, as depicted in Figure 8 in current research. Therefore, under the same slurry erosion test conditions, the volume loss of the three coating samples was significantly reduced, and the erosion corrosion performance was significantly improved.

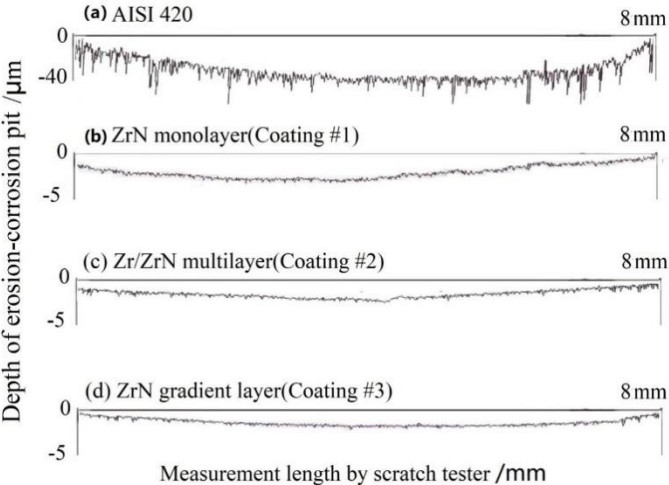

**Figure 12.** Surface profiles and curves of pits of samples after slurry erosion-corrosion tests.

In an erosion-corrosion process, the tangential motion of droplets and solid particles will be enhanced, while the normal motion will be weakened. In this way, the erosion-corrosion effects on the surface of samples are resulted from the synergistic effect of both the small-angle erosion of droplets and solid particles and the corrosion of liquid acid solution [36]. Previous studies have shown that Coatings #1, #2, #3 can significantly improve not only the erosion resistance of small-angle particles, but also the corrosion resistance of the substrate [37]. Therefore, the three coatings hinder the slurry erosion-corrosion behavior from above two aspects, thereby significantly improving the erosion-corrosion resistance of the AISI420 stainless steel [38]. In addition, compared with the other two coatings, the gradient layer (i.e., Coatings #3) has the best crack resistance and load-bearing capacity as well as good corrosion resistance. As such, the comprehensive results show that the erosion-corrosion damage of Coatings #3 is the smallest and its erosion corrosion resistance is the best.

## 5. Conclusions

(1) The microhardness of AISI420 stainless steel is measured to be 278 $HK_{0.1}$, while those of the ZrN monolayer, multilayer, gradient layer samples prepared by the ion-assisted arc deposition method are higher than 3000 $HK_{0.1}$. Among them, the ZrN gradient coating has the best crack resistance and bearing capacity, and its hardness decreases more slowly with an increasing test load.

(2) In acidic corrosive environments, ZrN monolayer, multilayer, and gradient coatings have significantly improved the corrosion resistance of the AISI420 stainless steel substrate, reducing its corrosion current density by 83.0%, 99.8%, and 95.8%, respectively. Because the ZrN multilayer and gradient layer coatings can constrain the growth of some columnar crystals and reduce the number of micro-pores inside the coating, their corrosion resistances are better than that of the ZrN monolayer coating.

(3) The ZrN coatings with three structures have significantly improved the bearing capacity and corrosion resistance of the AISI420 substrate, which further hinders the solid-liquid two-phase erosion-corrosion damage from the aspects of small-angle erosion resistance and corrosion resistance. Among them, the ZrN gradient coating has the best mechanical properties and electrochemical corrosion performance, and thus, it has the best erosion-corrosion resistance.

**Author Contributions:** Writing, Y.X., Investigation: L.W. (Lin Wan), Data curation: J.H. and Z.W., Writing—review & editing: L.W. (Lei Wang), Resources: D.Y., Project administration: D.L. Methodology: S.L. All authors have read and agreed to the published version of the manuscript.

**Funding:** This research was funded by Young Scientific Research and Innovation team of Xi'an Shiyou University (Grant No. 2019QNKYCXTD14), Provincial advanced subject of "Material Science and Engineering" in Xi'an Shiyou University, State Key Laboratory of Metastable Materials Science and Technology (Grant No. 202111), and State Key Lab of Advanced Metals and Materials (Grant No. 2021-Z06). The authors acknowledge a Collaborative Research and Development (CRD) Grant from the Natural Sciences and Engineering Research Council (NSERC) of Canada to D. Yang.

**Institutional Review Board Statement:** Not applicable.

**Informed Consent Statement:** Not applicable.

**Data Availability Statement:** Relevant data have been shown in the paper.

**Acknowledgments:** The present work has been financially supported by Young Scientific Research and Innovation team of Xi'an Shiyou University (Grant No. 2019QNKYCXTD14), Provincial advanced subject of "Material Science and Engineering" in Xi'an Shiyou University, State Key Laboratory of Metastable Materials Science and Technology (Grant No. 202111), and State Key Lab of Advanced Metals and Materials (Grant No. 2021-Z06). The authors acknowledge a Collaborative Research and Development (CRD) Grant from the Natural Sciences and Engineering Research Council (NSERC) of Canada to D. Daoyong Yang.

**Conflicts of Interest:** The authors declare no conflict of interest.

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
