# Peer review of "Improvement of Erosion-Corrosion Behavior of AISI 420 Stainless Steel by Ion-Assisted Deposition ZrN Coatings"

_metals, doi:10.3390/met11111811_

Round 1

Reviewer 1 Report

The article "Improvement of Erosion-Corrosion Behavior of AISI 420 Stainless Steel by Ion-Assisted Deposition ZrN Coatings" is well written and displays the entire range of tests needed to characterize the three types of coatings, from this point of view.

My comments and suggestions for minor changes are as follows:

Introduction: 

line 54,57. Please give full term for HVOF and WC, when first used.

line 58. Stellite 6 alloy. Please specify the type of alloy and manufacturer data.

line 71. corroded by the surrounding corrosive medium. Please avoid repeating.

Experiments:

line 98: PIEMAD-03 system. Please give manufacturer data.

Figure 1 shows a schematic diagram...Please rephrase, as Figure 1 not only shows a schematic diagram, but two images, as well.

Figure 1 caption. Please give specific information for a, b, c. The caption should be underneath the figure, not above. 

Same for Figure 2. 

line 136. HI-TACHIS-570. Please give manufacturer's data. Same for PARSTAT2273 (line 144).,  SY/Q-750 (line 149).

Results:

The parts mentioning: This observation is similar to those documented elsewhere [11, 24].-line 126; This is the same as those observed by Jiménez et al [25].- line 213; These scientific findings are consistent with those documented elsewhere [28].- line 241, should be extended with more information, or moved to the discussion section. 

The results should be summarized in tables, for better understanding, as you did in Table 2, for the erosion-corrosion test.

Author Response

Re: An Itemized Response to the Comments Made by Reviewer 1

Journal:    Metals

Manuscript No.: metals-1416338

Paper Title:        Improvement of Erosion-Corrosion Behavior of AISI 420 Stainless Steel by Ion-Assisted Deposition ZrN Coatings

Authors: Yuntao Xi, Lin Wan, Lei Wang, Hao Tong, Daoyong Yang, Daoxin Liu and Shubin Lei

Reviewer #1

  • Comment #1: In "Introduction:":

  1. i) line 54,57. Please give full term for HVOF and WC, when first used.
  2. ii) line 58. Stellite 6 alloy. Please specify the type of alloy and manufacturer data.

iii) line 71. corroded by the surrounding corrosive medium. Please avoid repeating.

Response:

  1. Yes, We have added the full name of HVOF and WC.
  2. Yes, we have pointed out the type of Stellite 6 alloy and manufacturer data.

iii) Yes, we have corrected this sentence.

Changes: Please see change in “introduction” part.

  • Comment #2: In "Experiments":

  1. i) line 98: PIEMAD-03 system. Please give manufacturer data.
  2. ii) Figure 1 shows a schematic diagram...Please rephrase, as Figure 1 not only shows a schematic diagram, but two images, as well.
    iii) Figure 1 caption. Please give specific information for a, b, c. The caption should be underneath the figure, not above. Same for Figure 2.

iv)line 136. HI-TACHIS-570. Please give manufacturer's data. Same for PARSTAT2273 (line 144).,  SY/Q-750 (line 149).

Response:

  1. We have added the manufacturer's information.
  2. Yes,  we re described this sentence and the corresponding graph caption.
  • Yes, we have modified the caption of Figures 1 and 2.
  1. We have added the corresponding information for these devices.

Changes: Please see change in “Experiments” part.

  • Comment #3: In "Results ":
  1. The parts mentioning: This observation is similar to those documented elsewhere [11, 24].-line 126; This is the same as those observed by Jiménez et al [25].- line 213; These scientific findings are consistent with those documented elsewhere [28].- line 241, should be extended with more information, or moved to the discussion section. 
  2. ii) The results should be summarized in tables, for better understanding, as you did in Table 2, for the erosion-corrosion test.

Response:

  1. In the result part, we describe the results and phenomena of the corresponding experiments and compare them with the similar results of previous studies, so we don't move the content here to the discussion part.
  2. We feed back the results of different experiments to the readers in the form of tables or graphs, which helps the readers to see and understand the experimental results intuitively and enrich the diversity of the results.

Sincerely yours,

Lei Wang and  Hao Tong (Corresponding author)

October 22, 2021

Reviewer 2 Report

The purpose of the manuscript is original and I found the topic quite interesting to read. The experimental methods are suitable to complete the characterization of the layers as well as to determine their corrosion resistance. I see, at some points, some lack of information, in the sense that some data that should be included in the manuscript are missed. From my view, the discussion is not well organized, authors must show firstly their findings, then try to explain them according to their results and finally look for other similar results in the bibliography to support their ideas.

List of corrections/comments:

  1. The authors’ details (just below the tittle) are indicated with a “letter” superscript but then they use numbers to refer to the individual institution
  2. What about the influence of the temperature? Blades will be subjected to very high temperatures, what is the influence of temperature on the coating performance? Did you perform any test to check this?
  3. Line 108, the number and the unit must be separated (check this along the manuscript)
  4. Figure 2c, correct the word “Govemor”, the correct is Governor
  5. Lines 129-131, concerning coating 2, what is the number of layers? 5? Then the total thickness is not 8 microns (5*0.9 + 5*1.2 = 10.5 microns)
  6. Line 176, image in figure 4b should be referred as the backscattered electron image (BSE)
  7. Line 189, authors say that the individual layers are 1.2 microns thick and 5 layers, right? This is what is observed in the pictures, but this coating stack is not clearly understood from your previous description in lines 129-131
  8. Figure 5c, the length reference bar is longer than for the other pictures (5a and 5b), is this picture taken at the same magnification? Also I do not see clearly the thickness of the coating in these three images, you could use the measurement tool available in the SEM (line scan) to determine exactly the thickness in these cross-section images (including these measurements in the images).
  9. Title in Figure 6, correct the format
  10. From the depth composition profile in figure 6, coating 3 is thicker than coating 1. You must consider in the manuscript these observed thicknesses instead of the expected thicknesses. What about the composition profile for coating 2?
  11. Figure 7, you must include the XRD results for all the coatings, even if they are the same, this would the evidence that, in fact, the surface composition is the same.
  12. Microhardness results, Figure 8 is not a microhardness profile, right? It is showing the variation of the hardness with the load; in a hardness profile you should include the several hardness values against the coating thickness. Did you try the hardness measurements in the cross section?
  13. How long were the salt fog spray tests? 120 hours (as written in the text) or 720 hours (included in figure 9)?
  14. Authors use this term “self-corrosion current density”, what is the meaning of self? This term is known as corrosion rate, or corrosion current density, but why do the authors use “self”?
  15. It is not clear how the erosion volume loss was obtained. Please include the procedure in the experimental description.
  16. Lines 286-289, authors wrote that some scratches are observed in the coated samples, but I can identify some pits (in the three coated samples), apparently the number of pits is smaller in coating 3, can you describe more in detail? The quality of the pictures, obtained after the erosion tests, for the coated samples is not as good as that for the uncoated sample (in figure 11)

Author Response

Re: An Itemized Response to the Comments Made by Reviewer 2

Journal:    Metals

Manuscript No.: metals-1416338

Paper Title:        Improvement of Erosion-Corrosion Behavior of AISI 420 Stainless Steel by Ion-Assisted Deposition ZrN Coatings

Authors: Yuntao Xi, Lin Wan, Lei Wang, Hao Tong, Daoyong Yang, Daoxin Liu and Shubin Lei

Reviewer #2

  1. i) The authors’ details (just below the tittle) are indicated with a “letter” superscript but then they use numbers to refer to the individual institution.
  2. ii) What about the influence of the temperature? Blades will be subjected to very high temperatures, what is the influence of temperature on the coating performance? Did you perform any test to check this?

iii) Line 108, the number and the unit must be separated (check this along the manuscript).

iv)Figure 2c, correct the word “Govemor”, the correct is Governor

  1. v) Lines 129-131, concerning coating 2, what is the number of layers? 5? Then the total thickness is not 8 microns (5*0.9 + 5*1.2 = 10.5 microns)
  2. vi) Line 176, image in figure 4b should be referred as the backscattered electron image (BSE)

vii) Line 189, authors say that the individual layers are 1.2 microns thick and 5 layers, right? This is what is observed in the pictures, but this coating stack is not clearly understood from your previous description in lines 129-131

viii) Figure 5c, the length reference bar is longer than for the other pictures (5a and 5b), is this picture taken at the same magnification? Also I do not see clearly the thickness of the coating in these three images, you could use the measurement tool available in the SEM (line scan) to determine exactly the thickness in these cross-section images (including these measurements in the images).

  1. ix) Title in Figure 6, correct the format
  2. x) From the depth composition profile in figure 6, coating 3 is thicker than coating 1. You must consider in the manuscript these observed thicknesses instead of the expected thicknesses. What about the composition profile for coating 2?
  3. xi) Figure 7, you must include the XRD results for all the coatings, even if they are the same, this would the evidence that, in fact, the surface composition is the same.

xii) Microhardness results, Figure 8 is not a microhardness profile, right? It is showing the variation of the hardness with the load; in a hardness profile you should include the several hardness values against the coating thickness. Did you try the hardness measurements in the cross section?

xiii) How long were the salt fog spray tests? 120 hours (as written in the text) or 720 hours (included in figure 9)?

xiv) Authors use this term “self-corrosion current density”, what is the meaning of self? This term is known as corrosion rate, or corrosion current density, but why do the authors use “self”?

  1. xv) It is not clear how the erosion volume loss was obtained. Please include the procedure in the experimental description.

xvi) Lines 286-289, authors wrote that some scratches are observed in the coated samples, but I can identify some pits (in the three coated samples), apparently the number of pits is smaller in coating 3, can you describe more in detail? The quality of the pictures, obtained after the erosion tests, for the coated samples is not as good as that for the uncoated sample (in figure 11)

Response:

  1. We uniformly use letters to represent the organization information in the manuscript. We will follow the requirements of the journal and make corresponding corrections.

  1. ZrN coating does not contain Ti and Crsingle film. It and TiN coating have high heat resistance, and are widely used in cutting tool and blade surface. The high temperature test results of the similar coatings show that the coatings have good properties at above 800 °C. 

References:IV Blinkov, DS Belov, AO Volkhonsky, et al. Heat Resistance, High-Temperature Tribological Characteristics, and Electrochemical Behavior of Arc-PVD Nano-structural Multilayer Ti–Al–Si–N Coatings[J],Materials Characterization, 2018, 140, 189-196.

  • Yes, we have made changes.

  1. Yes, we have made corrections.

  1. There may be some ambiguity in our expression. There are 5 layers, each with a thickness ranging from 0.9 microns to 1.2 microns. Each coating contains a 2.0 μm Zr primer layer. So the total of this coating is 8 microns.(5*1.2+2.0 = 8 microns) 

  1. Yes, we have made corrections.

  • Yes, the question has been revised in the text and answered in the questionv).

  • These three pictures are the same magnification; we have made correction about the length reference bar.The thickness of the three coatings can also be obtained from Fig.6, all about 8μm. 

  1. Yes, we have made corrections.

  1. From the depth composition profileanalysis, the thickness of the monolayer coating and the gradient coating is close to 8 microns, which is basically consistent with the predicted thickness. Since coating 3 and coating 1 are more representative in terms of coating hardness and corrosion resistance, so only their composition profiles are shown here.

  1. Because the three layers contain the same chemical composition, but the distribution is different, the XRD results of only one coating are listed.

  • Figure 8 is not the microhardness curve, but the change of hardness value of each coating and substrate with load. Because the coating is relatively thin, the hardness measurement of cross section is difficult to accurately reflect the hardness at positions with different thickness, so it is not carried out.

  • The salt spray experiment has been carried out for 120 hours, and we have corrected the clerical errors.

  • Yes, here is our clerical error. It should be corrosion rate, or corrosion current density, and we have corrected the clerical errors.

  1. Since the density of the substrate and each coating was different, the erosion process was evaluated according to the volume loss. Firstly, measure the volume of the sample before the erosion-corrosion test, and then measure the volume of the erosion pit caused by erosion-corrosion through the profiler to calculate the volume loss.

  • We have modified the corresponding picture and added a more detailed description.

Sincerely yours,

Lei Wang and  Hao Tong (Corresponding author)

October 22, 2021

Round 2

Reviewer 2 Report

There are still some minor mistakes that must be corrected:

The mistake concerning the superscript used along with the authors’ names and the organizations do not match (authors use a, b, c… for the authors and 1, 2, 3… for the organizations)

Line 112: 300 °C (revise carefully along the text)

Figure 11, the format used to describe each picture is different

Author Response

Re: An Itemized Response to the Comments Made by Reviewer 2

Journal:    Metals

Manuscript No.: metals-1416338

Paper Title:        Improvement of Erosion-Corrosion Behavior of AISI 420 Stainless Steel by Ion-Assisted Deposition ZrN Coatings

Authors: Yuntao Xi, Lin Wan, Jungang Hou, Zhiyong Wang, Lei Wang, Daoyong Yang, Daoxin Liu and Shubin Lei

Reviewer #2

  1. i) The mistake concerning the superscript used along with the authors’ names and the organizations do not match (authors use a, b, c… for the authors and 1, 2, 3… for the organizations).
  2. ii) Line 112: 300 °C (revise carefully along the text)

iii) Figure 11, the format used to describe each picture is different

Response:

  1. We have made changes to the author information.

  1. Yes, we have made changes.

  • Yes, we have made changeson the description of figures.

Sincerely yours,

Lei Wang  (Corresponding author)

October 28, 2021
